# Potential of Using Dual-Media Biofilm Reactors as a Real Coffee Industrial Effluent Pre-Treatment

Hassimi Abu Hasan [1,2,*], Dheenesh Sai Annanda Shanmugam [1], Siti Rozaimah Sheikh Abdullah [1,2], Mohd Hafizuddin Muhamad [1,*] and Setyo Budi Kurniawan [1]

[1] Department of Chemical and Process Engineering, Faculty of Engineering and Built Environment, Universiti Kebangsaan Malaysia, Bangi 43600, Selangor, Malaysia; p86663@siswa.ukm.edu.my (D.S.A.S.); rozaimah@ukm.edu.my (S.R.S.A.); setyobudi.kurniawan@gmail.com (S.B.K.)

[2] Research Centre for Sustainable Process Technology (CESPRO), Faculty of Engineering and Built Environment, Universiti Kebangsaan Malaysia, Bangi 43600, Selangor, Malaysia

* Correspondence: hassimi@ukm.edu.my (H.A.H.); ir_din84@yahoo.com.my (M.H.M.)

**Abstract:** The coffee processing industry produces toxic and low biodegradable effluent, which can pollute water bodies. A pre-treatment study on coffee effluent using a dual-media biofilm reactor (DM-BR) containing sand and Hexafilter (HEX) was conducted alongside a control biofilm reactor (C-BR) containing sand media. The novelty of this study lies in the use of dual media in biofilm reactor (DM-BR) for real coffee effluent treatment, where these processes were used individually in previous studies. The performance of DM-BR and C-BR in treating coffee effluent were investigated at different hydraulic retention times (HRTs), 24, 48 and 72 h, and the degrading bacteria were identified. Both biofilm reactors were inoculated with a recycled paper mill-activated sludge and acclimatised for 97 days. The DM-BR displayed the highest removal of chemical oxygen demand (COD) and $NH_4^+$-N at 47% and 38%, respectively, within 48 h of HRT, whereas colour and tannin–lignin reached maximum average removal of 21% and 29%, respectively, at 24 h of HRT. The combination of sand and HEX media in a system showed COD and $NH_4^+$-N removal improvement at 48 h of HRT and encouraged a variety of bacterial species growth. Bacterial characterisation analysis revealed *Proteobacteria* to be dominant.

**Keywords:** dual-media biofilm reactor; coffee effluent; aerobic degradation; lignin –tannin removal; cod removal





## 1. Introduction

Coffee processing consists primarily of two methods, specifically wet and dry methods, which are in contradiction with the complexity and quality of the resultant coffee [1]. The wet method is widely applied in coffee processing mills while it generates large amounts of effluent from the de-pulping, fermentation and washing steps in the process [2]. The uncontrolled release of these effluents is of great concern, as they contain high concentrations of suspended organics such as sugars, pectins, proteins and polyphenols and cause significant adverse effects on the receiving water bodies, such as the decreasing of dissolved oxygen that may disturb the aquatic ecosystem [3].

The effluent generated from this industry is acidic (pH 3–5). Additionally, it contains high concentrations of organic matter (1185–32,459 mg/L chemical oxygen demand (COD), 3450–12,100 mg/L biochemical oxygen demand ($BOD_5$), 7000–10,900 mg/L total suspended solids (TSS) and nutrients (4.4–70 mg/L phosphorus, 37–279 mg/L nitrogen) [4]. Furthermore, the effluent produced by coffee processing plants contains highly toxic and low biodegradability pollutants such as tannins, phenolic and alkaloids [5]. In addition to characteristics such as high toxicity and low biodegradability, the colour of the coffee effluent also poses a risk to the environment. Coffee wastewater has similar viscosity

and colour as very diluted black coffee [6]. This is due to the presence of dark brown pigments or melanoidins in these effluents, such as tannin and lignin. Once coffee effluent is discharged into waterways, it obstructs the light due to its extremely dark brown colour, thereby affecting photosynthesis. Moreover, eutrophication occurs as a result due to a high nutrient load. Significant pollution problems are also created due to the foul smell of coffee plant effluents [7]. These harmful effluents are released directly into the aquatic system, and can cause major health issues in humans, are toxic to aquatic life, and result in natural waters becoming unsuitable as potable water sources [8].

Thus, the current focus is to discover the most effective wastewater treatment method using low-cost sustainable technology to save the environment from the threat posed by coffee effluent [9]. In previous studies, various approaches were utilised to remove pollutants from coffee effluents, including zero-valent iron (ZVI) treatment [10], the photo-Fenton method [11], ultraviolet (UV) radiation catalysis (with ozone) [12], electro-oxidation [13], membrane filtration [14,15], chemical coagulation and flocculation [16]. However, due to the high footprint requirement, high energy consumption and huge operation and maintenance costs [17], most of their use is restrained and requires steady financial input. In addition, to the best of our knowledge, lack of study using biological approaches for the coffee effluent treatment. Laboratory scale granular sludge bed showed removal of COD up to 94% from coffee wastewater. In addition, coffee biodegradability was demonstrated in the laboratory using a complete stirred tank reactor, achieving >60% degradation under anaerobic conditions. Due to the limited research on the biological treatment of coffee effluent, there is still a need to develop an economically viable and eco-friendly technology for handling such types of wastewaters.

Most of the water usage in coffee mills is unsupervised, especially the amount of water used in the washing line. As a result, a superfluous amount of water is used during the process. A previous study revealed that 87% of coffee mills in Costa Rica do not supervise water usage aside from the amount of water used for the main process [18]. A coffee processing mill in Penang, Malaysia, produces a substantial amount of coffee effluent originating from the washing line. This coffee wastewater is highly acidic. Although these coffee effluents are subjected to a rapid sand filtration system, they still contain high amounts of COD and have high turbidity, more than the permissible levels as stated in Standard B of the Environmental Quality (Industrial Effluent) Regulations 2009 [19].

Immobilization and growth of biomass as biofilms is proved to be a promising method to retain slow-growing microorganisms in bioreactors that operate continuously [20]. Supporting media provide a high surface area for biofilm development, which results in less sensitivity to toxic compounds [21]. Previous studies showed that supporting media, e.g., polyethylene plastics, polyvinyl alcohol gels, sand, granular activated carbon, polymer foam pads and polyurethane sponges, have been introduced to the biofilm process [22]. However, to our concern, data related to the use of multi-type supporting media in biofilm reactor were scarce, as the concept of multi-type supporting media were considered new compared to the single-type biofilm carrier.

Therefore, in this study, we developed a dual-media biofilm reactor (DM-BR) as a pre-treatment system, containing a combination of submerged and floating media, sand and a Hexafilter (HEX), respectively, to treat the coffee effluent. The novelty of this study lies in the use of dual media in a biofilm reactor (DM-BR) for real coffee effluent treatment, which has not been reported thus far. In terms of system design, the sand media in the DM-BR serves as a filter preventing biomass washout and providing a large surface area for faster biofilm development. Increased stability and performance in the DM-BR can be achieved if the microbial consortium is retained in the reactor. As the sand media on which microorganisms grow is fluidised state, the surface of the media available for the development of microorganisms is quite large, which leads to a high concentration of microorganisms. Because of a large concentration of microorganisms, DM-BR bears a high potential for the removal of various parameters such as COD, nitrogen, colour, etc. If DM-BR is operated properly, there is no need to provide a secondary setting tank, which

leads to a saving in the total cost of the plant. In addition to sand media, HEX also acts as an effective support media for the growth of slimy biofilm to increase the overall performance of the system.

Thus, this biofilm reactor has the potential as a simple, reliable, inexpensive and environmentally friendly alternative [22–25] compared to the existing conventional rapid sand filters used in coffee processing mills. In addition to the DM-BR, a control biofilm reactor (C-BR) containing sand media was developed. Both biofilm reactors were inoculated with activated sludge from a recycled paper mill as a biofilm source to treat coffee effluent pollutants. In this study, we investigated the performance of both biofilm reactors in treating coffee effluent and subsequently identified the coffee effluent degrading bacteria.

## 2. Materials and Methods

### 2.1. Sampling of Coffee Industry Effluent

Coffee industry effluent (CIE) was sampled from a coffee processing factory located in Tasek Gelugor, Penang, Malaysia. The CIE was produced from the washing wastewater and water from the scrubber process. The effluent was preserved in a cool room at 4 °C prior to use. The characteristics of the coffee effluents are shown in Table 1. The CIE had high amounts of COD, $NH_4^+$-N and colour. The CIE consisted of macromolecules of tannin–lignin, which contributed to the colour of the effluent. The CIE was acidic with a pH of 4.6. The COD, $NH_4^+$-N, pH, MLSS and colour levels in the CIE exceeded the Standard B regulated by the Department of Environment Malaysia. Due to the high strength of the CIE, with a COD concentration of greater than 15,000 mg/L, the effluent was diluted with distilled water in accordance with a dilution factor of 0.1 (*v/v*) to avoid the biological aerobic treatment being affected [26] and to simulate as a primary treated CIE prior to treatment using biofilms process.

**Table 1.** Coffee effluent characteristics.

| Parameter | Coffee Effluent Mean Values | | Standard B (Federal Subsidiary Legislation Malaysia 2009) |
|---|---|---|---|
| | **Raw** | **10% Coffee + 90% Water** | |
| COD (mg/L) | 15,700 | 900–1050 | 200 |
| $NH_4^+$-N (mg/L) | 93 | 4.5–8.5 | 20 |
| pH | 4.6 | 4.5 | 5.5–9.0 |
| MLSS (mg/L) | 123 | <10 | 100 |
| MLVSS (mg/L) | 111 | - | - |
| Turbidity (NTU) | 74.7 | 15–20 | - |
| Colour (ADMI) | 1867 | 1020–1100 | 200 |
| Tanin–Lignin (mg/L) | 510 | 35–40 | - |

### 2.2. Bacterial Inoculation and Acclimatisation

Activated sludge from an industrial wastewater treatment plant located in Kajang, Malaysia, was used as a bacterial seed. Approximately 20% of the activated sludge with an initially mixed liquor suspended solids (MLSS) and mixed liquor volatile suspended solids (MLVSS) for DM-BR and C-BR ranged from 100–200 mg/L and 90–150 mg/L, respectively. Bacterial acclimatisation and development were conducted for DM-BR and C-BR using synthetic wastewater for 97 days (3 months). The reactors were initially fed with 3 L of synthetic wastewater containing glucose ($C_6H_{12}O_6$, 200 mg/L) and ammonium chloride ($NH_4Cl$, 40 mg/L). Without any addition of trace elements, only $C_6H_{12}O_6$ and $NH_4Cl$ as a carbon and nutrient source, respectively, were supplied to promote the growth and propagation of the bacteria. Concentrations of $C_6H_{12}O_6$ and $NH_4Cl$ were increased gradually to 500–1000 mg/L and 50–90 mg/L, respectively. The acclimatisation was conducted at an HRT of 24 h with DO and pH values maintained at 2–7 mg/L and pH 6–8, respectively. During the acclimatisation, the main wastewater quality parameters such as COD, $NH_4^+$-N, pH, DO, MLSS and MLVSS were monitored.

### 2.3. Configuration of the Dual-Media Biofilm Reactor

#### 2.3.1. Biofilm Carriers

HEX was made from polyethene, with a diameter and height of 25 mm and 12 mm, respectively. It consisted of a large surface and specific surface area of 1460 mm$^2$ and 320 m$^2$/m$^3$, respectively. The specific gravity of 1.03 allowed for it to float on the effluent surface. The HEX that floats and is continuously in motion inside CIE acts as a biofilm attachment site due to its large surface area. Meanwhile, the sand with a granule size of 0.5–1.18 mm was used as fixed carriers.

#### 2.3.2. Setup and Operation of the Biofilm Reactors

Two laboratory reactors were fabricated using Plexiglas with a dimension of 19 cm (D) × 25 cm (H), as shown in Figure 1. Figure 1a shows DM-BR filled with sand (fixed carrier) + HEX (moving carrier), while Figure 1b shows C-BR filled with sand (fixed carrier). The working volume of both reactors was 3 L. The sand carriers were filled at the bottom of both reactors to a height of 8 cm with a filling ratio of 30%. The HEX (110 pieces) was filled into another reactor at 25% of the reactor working volume. Air distributors were placed at the bottom of the reactors, and aeration was supplied using an aquarium pump (Model BB 8000, Aqualeisure, China) to ensure adequate oxygen supply for bacterial aerobic activity.

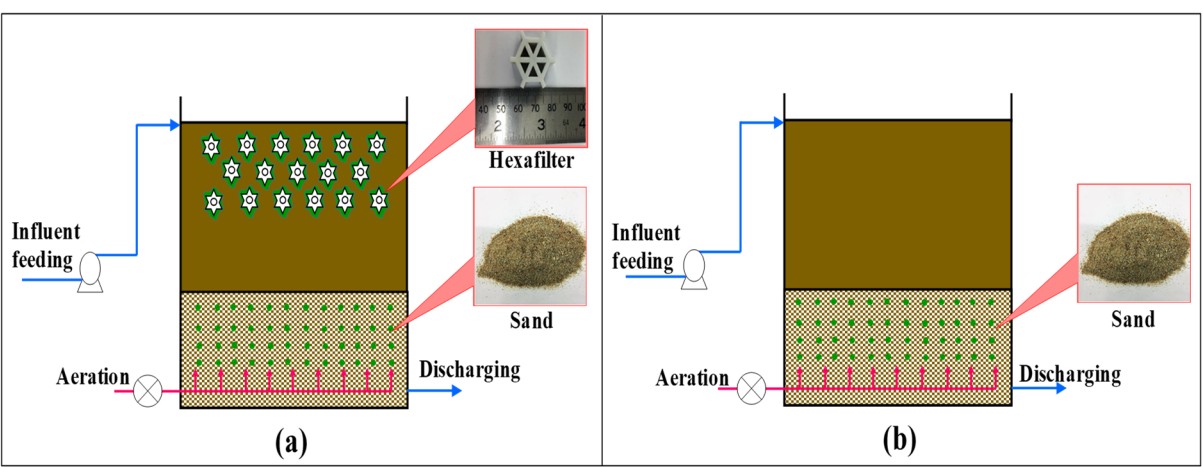

**Figure 1.** Schematic diagram of (**a**) DM-BR and (**b**) C-BR.

Both reactors were operated in a batch mode with a sequencing method of 0.25 h filling, 23–71 h reaction, 0.25 h settling and 0.5 h discharging. The discharged effluent from both reactors was 80%. The HRT varied from 24 to 72 h with loading rates of 13.5–40.6 mg/L/h. The performance of both reactors in treating CIE under various HRTs was monitored for 76 days, excluding the acclimatisation period. The acidic CIE was neutralised with sodium carbonate to avoid inhibiting bacterial growth. Due to the high strength of the CIE, the effluent was diluted at a ratio of 1:9 (CIE–water). The treatment was started with an HRT of 24 h (Days 97–118), which was then increased to an HRT of 48 h (Days 119–140) and 72 h (Days 141–173). The treated CIE was sampled at the end of each HRT prior to analysis.

### 2.4. Wastewater Quality Parameter Analysis

Effluent samples from both reactors were collected in plastic bottles at the end of each HRT. A glass microfibre filter (0.45 μm pore size; Whatman, Florham Park, NJ, USA) was used to filter the excessive suspended solids via a gravimetric method. The samples used for tannin–lignin concentration were examined using the Tyrosine Method (Method 8193, HACH, Loveland, CO, USA). Samples for COD measurement were digested in a digestion reactor (Method 8000, HACH). NH$_4^+$-N concentrations were analysed via the Nesslerization method (Method 8038, HACH). Colour was examined using the ADMI Weighted Ordinate Method (Method 10048, HACH). Measurements of tannin–lignin, COD,

$NH_4^+$-N and colour were performed using a HACH DR6000 Spectrophotometer (Loveland, CO, USA). MLSS and MLVSS in the suspension and attached to the carriers were examined via the gravimetric method and were dried at 105 °C and 550 °C, respectively [27]. pH was measured using a pH meter (Metrohm 827, Herisau, Switzerland) while DO levels were determined using a DO meter (Model YSI 550A, Yellow Springs, Ohio, USA). Samples for turbidity concentrations were evaluated using a turbidity meter (HACH 2100AN, Loveland, CO, USA). The equation used to calculate the removal efficiency (%) in the experiments was:

$$R\ (\%) = \frac{C_0 - C}{C_0} \times 100 \tag{1}$$

where $C_0$ and $C$ are the initial and final concentrations (mg/L), respectively.

### 2.5. Bacterial Analysis

2.5.1. Biofilm Observation under Scanning Electron Microscopy

The activated sludge suspension and biofilm formation on the sand and HEX surfaces were observed under scanning electron microscopy (SEM, Zeiss Supra 55VP, Germany). The samples were fixed with 4% glutaraldehyde for 12–24 h at 4 °C, followed by washing three times with 0.1 M phosphate buffer (10 min each). The samples were then washed with 35%, 50%, 75% and 95% acetone, and each washing lasted approximately 10 min. Finally, the samples were washed three times with 100% acetone (approximately 10 min each). The samples were subsequently transferred to a Critical Point Dryer for 30 min before being placed onto a stub using two-sided bands or colloidal silver. Finally, the samples were gold-plated with sputter plugs prior to being viewed in the SEM.

2.5.2. Microbial Cell Number

The microbial cell number was determined for biomass suspension at the end of acclimatisation (day 97) and treatment (day 173). It was determined using the plate count method by spreading the biomass suspension at a few serial dilutions [28,29]. The estimation number of viable bacteria on the plate was presented as colony-forming unit (CFU) per mL sample.

2.5.3. Microbial Community Analysis

The microbial community structure was analysed via DNA extraction, PCR amplification, PCR clean-up and quantification, and Illumina sequencing. Microbial DNA was extracted from the biofilm sample taken from the HEX using a FavorPrep™ Soil DNA Isolation Mini Kit (Favorgen, Taipei, Taiwan) according to the manufacturer's protocol. During the PCR amplification process, the regions of bacterial DNA from V3 to V4 were amplified at 95 °C for 3 min followed by 25 cycles at 95 °C for 30 s, 55 °C for 30 s and 72 °C for 5 min using the following primer sequences:16S Amplicon PCR Forward Primer = 5' TCGTCG-GCAGCGTCAGATGTGTATAAGAGACAGCCTACGGGNGGCWGCAG and 16SAmplicon PCR Reverse Primer = 3' GTCTCGTGGGCTCGGAGATGTGTATAAGAGACAGGAC-TACHVGGGTATCTAATCC. PCR reactions were performed in triplicate and involved 25 μL of 2 × KAPA Hifi HotStart ReadyMix (12.5 μL), 5 μL amplicon PCR forward primer of each primer (1 μM), 5 μL amplicon PCR forward primer of each primer (1 μM), and 2.5 μL of microbial DNA (5 ng/μL). The amplified DNA was purified using PCR clean-up kits containing 10 mM Tris at pH 8.5, 80% ethanol and AMPure XP beads according to the manufacturer's protocol. A clean-up step followed to purify the 16S V3 and V4 amplicons from the free primers and primer-dimer species. Lastly, in the Illumina sequencing, sample libraries were pooled in equimolar concentrations and were paired-end sequenced using the Illumina MiSeq platform according to standard protocols.

Chao and ACE analysis were used to estimate the number of species in a biomass sample. Chao was calculated using Equation (2) as follows:

$$S_{est} = S_{obs} + \frac{f_1(f_1 - 1)}{2(f_2 + 1)} \tag{2}$$

where, $S_{est}$ = estimated richness, $S_{obs}$ = observed number of species, $f_1$ = number of operational taxonomic units (OTUs) with only one sequence and $f_2$ = number of OTUs with only two sequences [30].

Shannon, Simpson and Fisher analyses were utilised to show the diversity index of the microbial community in which the index increased as the richness and evenness of the microbial community increased. Equations (3)–(5) were used to calculate the Shannon, Simpson and Fisher diversity indices, respectively.

$$H' = \sum_{i=1}^{R} p_i \ln p_i \tag{3}$$

$$D = \sum \left(\frac{n}{N}\right)^2 \tag{4}$$

$$S = a \ln\left(1 + \frac{n}{a}\right) \tag{5}$$

where $R$ = number of observed OTUs, $p_i$ = proportion of species, $i$ = relative to the total number of species, $n$ = total number of organisms of a particular species, $N$ = total number of organisms of all species, $S$ = number of taxa and $a$ = Fisher's alpha [31].

## 3. Results and Discussion

### 3.1. Bacterial Enhancement and Acclimatisation

Enhancement and acclimatisation of bacteria were conducted for 3 months. As summarised in Table 2, during the initial acclimatisation (days 0–14), the reactors were fed with 200 mg/L COD and 40 mg/L $NH_4^+$-N. The bacteria consumed all the organic carbon (COD = 100% removal) but not much of the nitrogen ($NH_4^+$-N = 12.4% removal). Between days 16 and 64, the COD concentration was increased to 500 mg/L to ensure the bacteria had sufficient organic carbon supply, while the concentration of $NH_4^+$-N was maintained. In accordance with the aforementioned concentration of COD and $NH_4^+$-N, the removal of COD decreased to 88.8%; meanwhile, $NH_4^+$-N removal recorded was only less than 5%. Low removal of $NH_4^+$-N was observed via the accumulation of a black layer in the reactor, which contained a sulphite precipitate that disrupted the nitrogen consumption by the bacteria [32]. The use of ammonium sulphate during the initial acclimatisation caused this issue as sulphides are generated from dissimilatory sulphate or sulphur reduction. Hence, the nitrogen source was replaced with $NH_4CI$ after day 64. From day 65 to 77, the COD removal increased to 91.6%, while $NH_4^+$-N also drastically increased to 38.8%. By increasing the concentration of glucose and $NH_4CI$ at the end of the acclimatisation period (days 78–97), the removal of COD and $NH_4^+$-N for both reactors was approximately 90% and 40%, respectively. The MLSS and MLVSS at the end of the acclimatisation period for both reactors increased to 1750–1900 mg/L and 1650–1680 mg/L, respectively. The pH and DO were observed in the range of 6–9 and 6–7 mg/L, respectively.

**Table 2.** Removal of COD and $NH_4^+$-N in DM-BR and C-BR throughout acclimatisation.

| Day | Influent (mg/L) | | Average Removal (%) | | | |
|---|---|---|---|---|---|---|
| | | | C-BR | | DM-BR | |
| | COD | $NH_4^+$-N | COD | $NH_4^+$-N | COD | $NH_4^+$-N |
| 0–15 | 200 | 40 | 100 | 12.4 | 100 | 10 |
| 16–64 | 500 | 40 | 88.8 | 4.4 | 88.6 | 1.2 |
| 65–77 | 600–800 | 50–70 | 91.6 | 38.8 | 92.3 | 34.6 |
| 78–97 | 1000 | 80–90 | 89.6 | 39.7 | 89.3 | 44.2 |

*3.2. Performance of the Biofilm Reactors*

3.2.1. Removal of Tannin–Lignin

The dark brown colour of the CIE is due to the organic recalcitrant compound of tannin and lignin. The performance of tannin–lignin removal is shown in Figure 2. The compound showed a slow degradation rate and could be harmful to the environment. The average tannin–lignin removal at an HRT of 24 h was 28.8% (Table 3), with an average effluent concentration of 25.8 mg/L. At an HRT of 48 h, the average tannin–lignin removal decreased slightly to 26.4% (Table 3), with an average effluent concentration of 28.6 mg/L. Meanwhile, at an HRT of 72 h, the removal of tannin–lignin was similar to an HRT of 24 and 48 h. This finding suggests that an HRT from 24 to 72 h insignificantly affected tannin–lignin removal in the dark brown CIE. Tannin–lignin compounds have complex molecules with a high molecular weight and are persistent towards degradation due to the presence of carbon-to-carbon linkage of biphenyl type and other linkages in the molecule [33]. The degradation of tannin–lignin has also been studied in pulp and paper wastewater using granular sludge technology, where the removal decreased at high concentrations [34]. Research conducted by Diez et al. [35] on kraft mill wastewater treatment using activated sludge showed that at an HRT of 10 h, the removal of tannin–lignin was in the range of 25–48%. Another study on CIE reported that lignin removal remained unchanged with a range of 9–11% for mesophilic and thermophilic anaerobic digesters [36]. The findings of this study revealed that the removal of high molecular weight compounds such as tannin–lignin is very challenging with the use of aerobic biofilms or anaerobic treatments.

**Table 3.** Summary of CIE treatment performance by DM-BR and C-BR.

| Reactor | Day | HRT (h) | Average Removal (%) | | | | |
|---|---|---|---|---|---|---|---|
| | | | Tannin–Lignin [a] | COD [a] | $NH_4^+$-N [a] | Colour [a] | Turbidity [a] |
| DM-BR | 97–118 | 24 | 28.8 ± 18.5 | 40.9 ± 10.9 | 10.4 ± 1.2 | 21.1 ± 19.2 | 21.2 ± 23.1 |
| | 119–140 | 48 | 26.4 ± 9.6 | 47.0 ± 4.2 | 37.6 ± 0.6 | 6.7 ± 2.1 | 32.7 ± 12.3 |
| | 141–173 | 72 | 26.3 ± 7.9 | 44.2 ± 5.2 | 12.3 ± 2.9 | 5.4 ± 4.9 | 23.8 ± 18.7 |
| C-BR | 97–118 | 24 | 23.3 ± 15.7 | 37.9 ± 9.2 | 0 | 10.0 ± 1.6 | 14.0 ± 18.9 |
| | 119–140 | 48 | 24.0 ± 16.9 | 42.2 ± 3.7 | 23.4 ± 19.7 | 4.7 ± 3.6 | 46.0 ± 21.4 |
| | 141–173 | 72 | 25.4 ± 8.4 | 38.7 ± 16.9 | 12.5 ± 19.2 | 4.6 ± 4.3 | 32.8 ± 25.2 |

[a] Values are means ± standard deviations based on total samples collected throughout treatment periods (days).

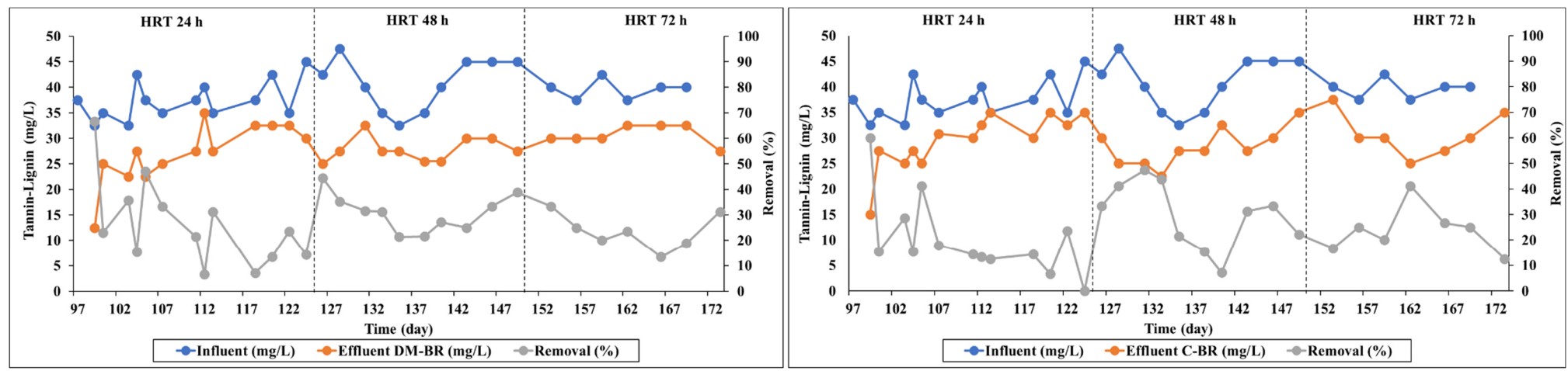

**Figure 2.** Removal of tannin–lignin in the DM-BR (**left**) and the C-BR (**right**).

3.2.2. Removal of COD and $NH_4^+$-N

Figure 3a shows the performance of COD removal for HRTs of 24 to 72 h. The average influent of COD in diluted CIE was 969.8 mg/L. No significant change in COD removal was observed with an increase in HRT. At an HRT of 24 h, 40.9% COD removal was achieved (Table 3) with an average effluent concentration of 585.8 mg/L. An increase in the HRT to 48 h showed a slight increase in the removal with an average removal of 47% (Table 3) with an average effluent concentration of 535 mg/L, while at an HRT of 72 h with a lower feeding concentration, the removal was 44.2% (Table 3). The performance of DM-BR dropped compared to during the acclimatisation stage. This suggests that organic compounds in coffee effluent are refractory toward aerobic biofilm degradation. Figure 3b shows the removal performance of $NH_4^+$-N. As shown in Table 3, the most effective HRT for $NH_4^+$-N removal was achieved at 48 h, with removal at 37.6%. At HRTs of 24 and 72 h, the removal was 10.4% and 12.3%, respectively. A C/N ratio of 160 contributed to the imbalanced treatment performance, while previous research mentions a lower C/N ratio of 137 for bioreactors [37]. A high C/N ratio might result in lower degradation performance since organic materials (especially carbon) are consumed and converted into cell biomass under a specific ratio [38].

Extensive previous studies revealed that the quality of existing carbon sources in real industrial effluents was poor and could not be consumed by microorganisms, which further worsened the degradation process. There was greater removal of COD and $NH_4^+$-N during the acclimatisation stage due to the availability of an easily degraded carbon source. In contrast, real CIE contains degradable and non-degradable carbon sources, where non-degradable carbon sources contribute to the low removal performance of the DM-BR.

3.2.3. Removal of Colour and Turbidity

Figure 4a shows the average removal of colour. Colour due to the presence of lignin or polymerised tannins in the CIE showed a poor biodegradable rate as the HRT increased. The highest removal was achieved at an HRT of 24 h with an average removal of 21.1% (Table 3); however, the reactor did not function well to remove the colour even when the HRT was increased to 48 and 72 h. The colour, which was contributed to by the presence of the macromolecules of tannin–lignin, required multiple physical and chemical treatments to achieve high removal [39]. The trend of colour removal was correlated with the tannin–lignin concentration due to the carbon-carbon double bond of tannin–lignin undergoing cleavage and degrading to $CO_2$ and $H_2O$, thus simultaneously removing the colour in the coffee effluent [7]. Figure 4b shows the turbidity removal throughout the treatment period. The influent turbidity concentration ranged from 14 to 30 NTU. It was found that the removal of the turbidity was in the range of 21.2–23.8%, with the highest removal achieved at 48 h HRT with 32.7% removal.

*3.3. Monitoring of Operational Parameters*

DO is an important factor in the biological degradation process for both COD and $NH_4^+$-N removal. Throughout the study, the average DO concentration in the influent was 3.5 to 4.5 mg/L and was 5 to 6 mg/L in the effluent. In addition, the average pH value was from 6 to 7. The DM-BR retained bacterial cells in a biofilm that adhered to the sand and HEX media surface as well as being suspended in the reactor. Throughout the treatment phase, the average MLSS and MLVSS ranged from 400 to 2700 mg/L and from 400 to 2300 mg/L, respectively. The mixed liquor increased from time to time until the end of the treatment, indicating the enhancement of microbial cells. The high surface area of the hybrid media (sand and HEX) in the reactor allowed for ease of adherence for the bacteria.

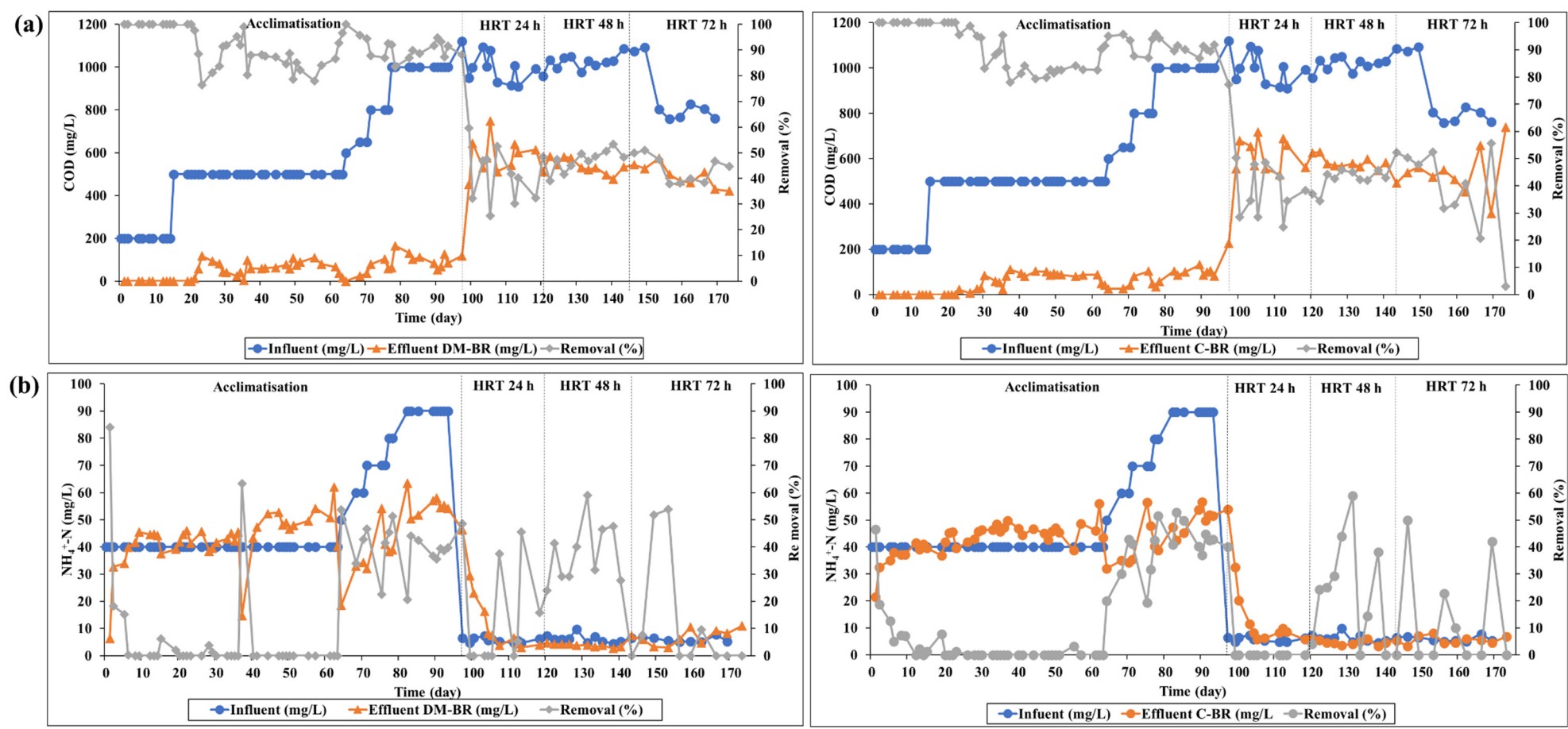

**Figure 3.** Removal of (**a**) COD and (**b**) NH4+-N in the DM-BR (left) and the C-BR (right).

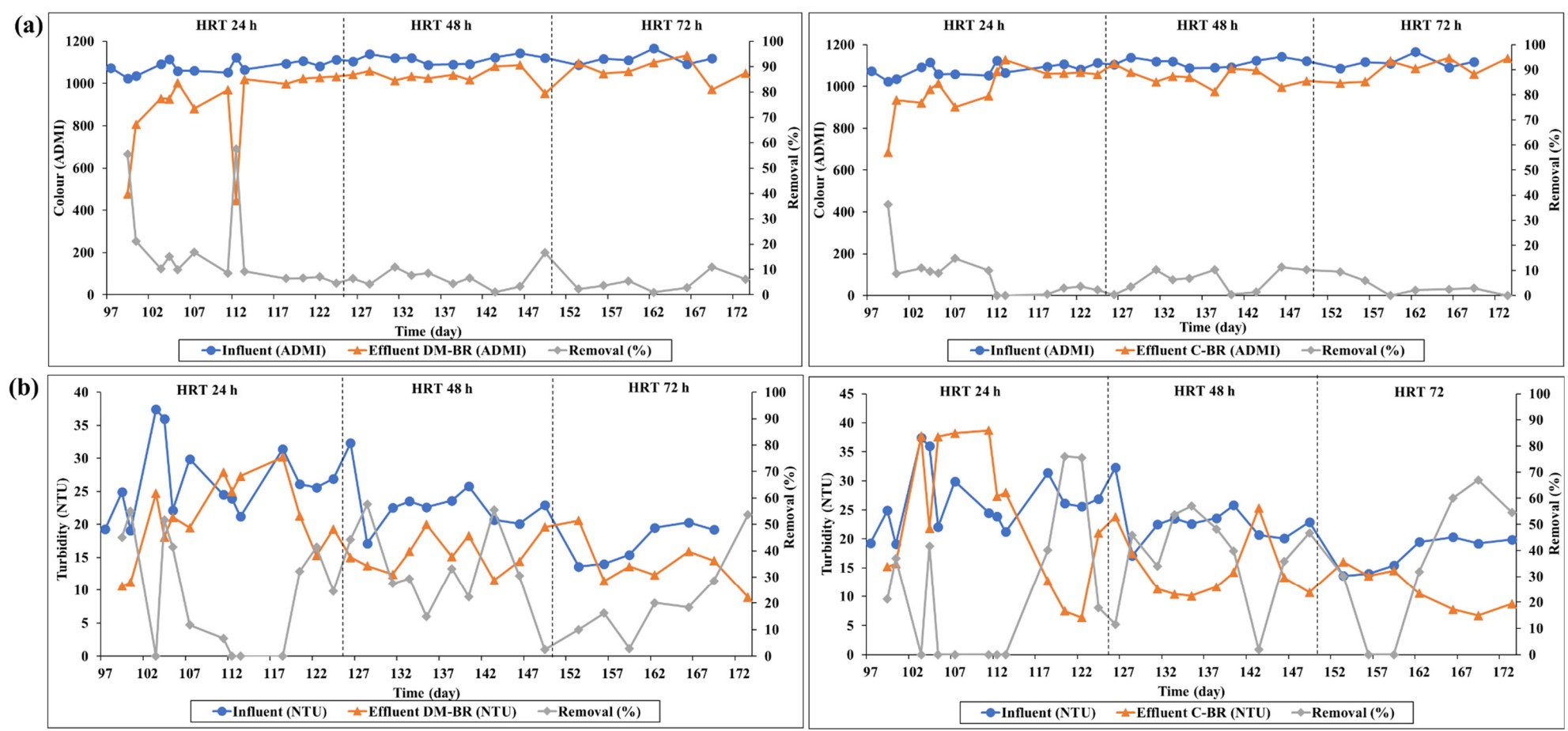

**Figure 4.** Removal of (**a**) colour and (**b**) turbidity in the DM-BR (left) and the C-BR (right).

### 3.4. Microbial Community Structure and Analysis

3.4.1. Microbial Observation via SEM

SEM analysis was conducted after the treatment for samples of biomass attached on HEX and sand and suspended biomass, as shown in Figure 5. As can be seen from the micrograph, bacterial biofilms are complex consortia of cells and extracellular polymeric substances (EPS). The micrographs results were used to classify the bacteria as coccus-shaped, rod-shaped and EPS matrixes. At the end of the treatment, these consortia of microorganisms formed complex granular and rough surfaces where the rough surfaces increased the number of effective sites for pollutant removal. As portrayed in the micrographs, the biofilms formed a layered structure (growth gradient). This occurred because of diffusional substrate concentration gradients. Fast-growing bacteria (heterotrophic) tend to grow outside the biofilm matrix, while low-growing bacteria (autotrophic) develop inside the biofilm so that it remains protected from the external sheer force, which could lead to detachment [40]. Coccus-shaped bacteria were rarely observed compared to rod-shaped bacteria, which were observed inside the biofilm matrix, indicating that coccus-shaped bacteria were autotrophic and were responsible for the removal of $NH_4^+$-N from the coffee effluent.

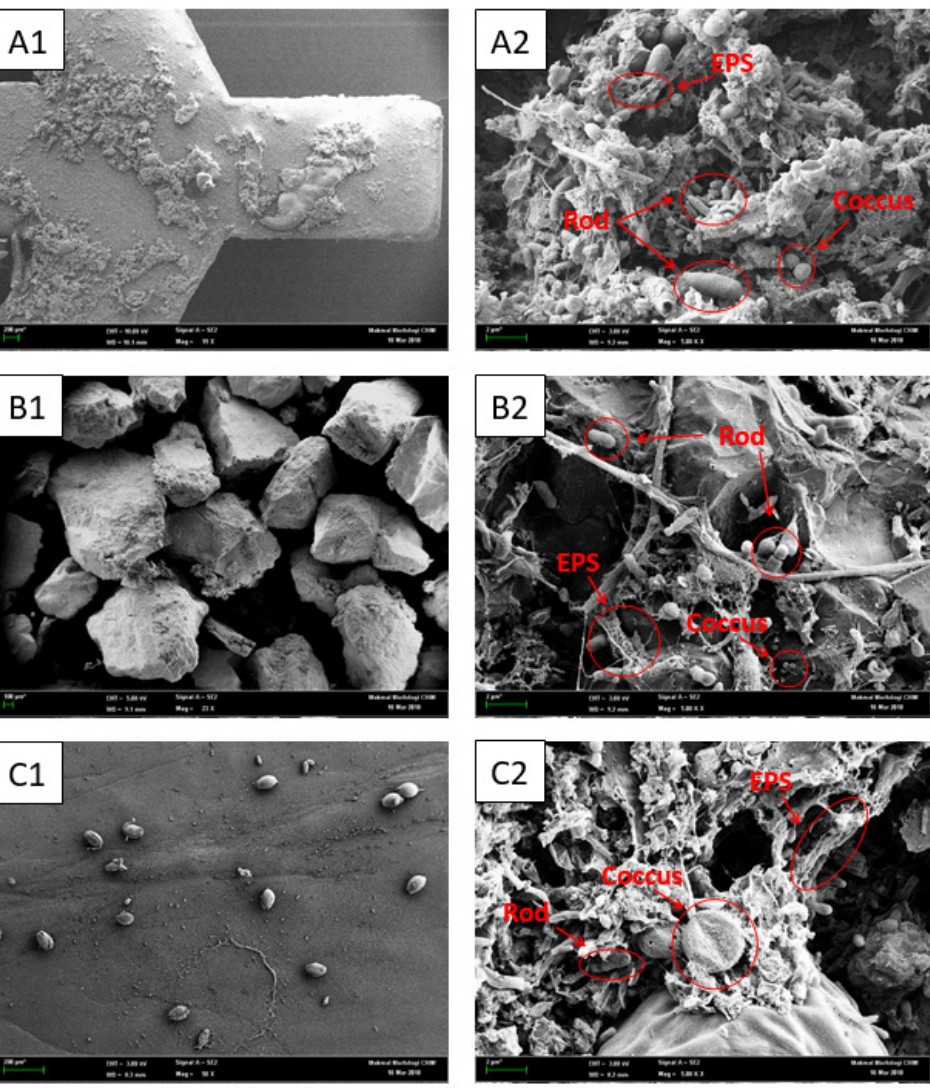

**Figure 5.** SEM micrographs for (**A**) Biofilm attached on HEX media: A1 at magnificent 19X, A2 at magnificent 5 k X, (**B**) biofilm attached on sand: B1 at magnificent 23X, B2 at magnificent 5 k X, and (**C**) suspended biomass in coffee mill effluent: C1 at magnificent 50X, C2 at magnificent 5 k X.

### 3.4.2. Microbial Cell Number

The total bacteria count at the end of the acclimatisation period before treatment (day 97) and after treatment (day 173) is shown in Figure 6. Prodigious colonies were observed for HEX at the end of the treatment. HEX was the most effective media for biofilm growth as it had the greatest number of bacterial cells ($1.42 \times 10^8$ CFU/mL) compared to any other samples (after treatment) due to a large active surface area for biofilm development. The number of bacterial cells on the sand and in suspension in the coffee wastewater decreased at the end of the treatment. A similar trend was observed for the total cell number in the reactor. This could be due to the toxicity effect of the real coffee effluent on the bacterial cells.

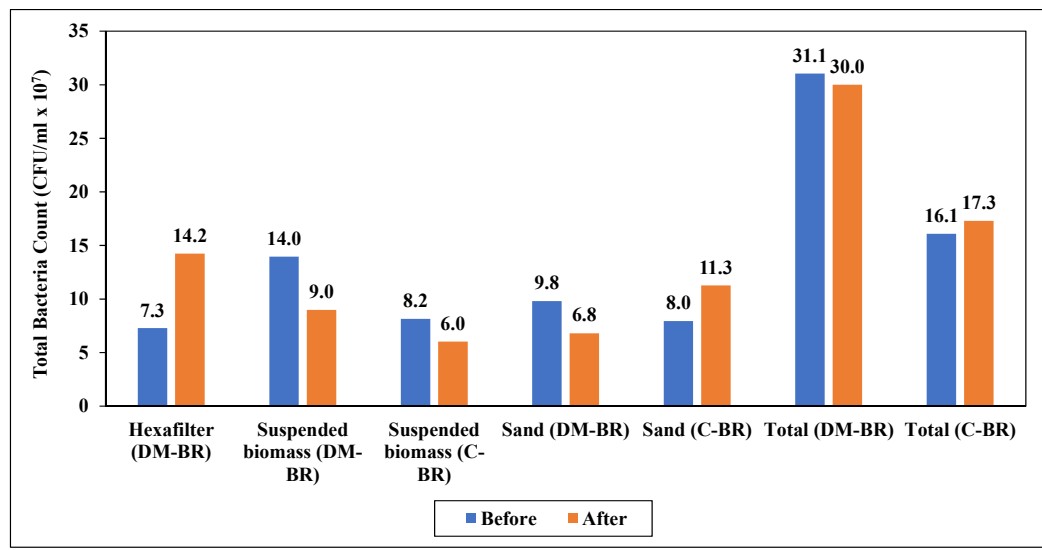

**Figure 6.** Total bacterial count for samples before (day 97) and after treatment (day 173).

### 3.4.3. Microbial Community Diversity

Metagenomic analysis of the microbial community revealed a diverse microbial presence in the reactor. Table 4 depicts the diversity parameters, including Chao, ACE, Shannon, Simpson and Fisher indices. All diversity parameters excluding Fisher's index showed the richness of the microbial community compared to the C-BR in which sand was utilised as the sole biofilm carrier. The Chao1 and ACE values were 3051 and 24, respectively. Meanwhile, the diversity indices represented by Shannon and Simpson were 24.1 and 6.0, respectively. The richness of diversity in the DM-BR containing the sand + HEX indicates that a high surface area for attachment led to the richness of the microbial community, thus allowing for the high efficiency of CIE treatment.

**Table 4.** The diversity of microbial in the DM-BR.

| Diversity Parameters | DM-BR | C-BR |
|---|---|---|
| OTU | 3641 | 3359 |
| Chao1 | 3051 | 2941 |
| ACE | 24 | 21 |
| Shannon | 24.1 | 22.8 |
| Simpson | 6.01 | 5.9 |
| Fisher | 801 | 99 |

An analysis of the microbial community which adhered to the sand + HEX during the CIE treatment revealed a host of organisms that were classified into phylum and genus, as shown in Figure 7. Figure 7a depicts 12 phyla where *Proteobacteria* was the most dominant phylum for the DM-BR. Figure 7a also shows that both reactors were dominated by the phyla *Proteobacteria*, *Planctomycetes*, *Verrucomicrobia*, *Actinobacteria*, *Acidobacteria*, *Chlamydiae*

and *Spirochaetes*. Figure 7b shows the genus level of the microbial community. The most abundant genera in the C-BR were *Treponema*, *Devosia*, *Candidatus Rhabdochlamydia*, *Geothrix*, *Mycobacterium* and *Rhodoplanes*. The most abundant genera in the DM-BR consisted of *Opitutus*, *Dok59*, *Burkholderia*, *Nitrospira*, *Mycobacterium*, *Rhodoplanes* and *Gemmatimonas*.

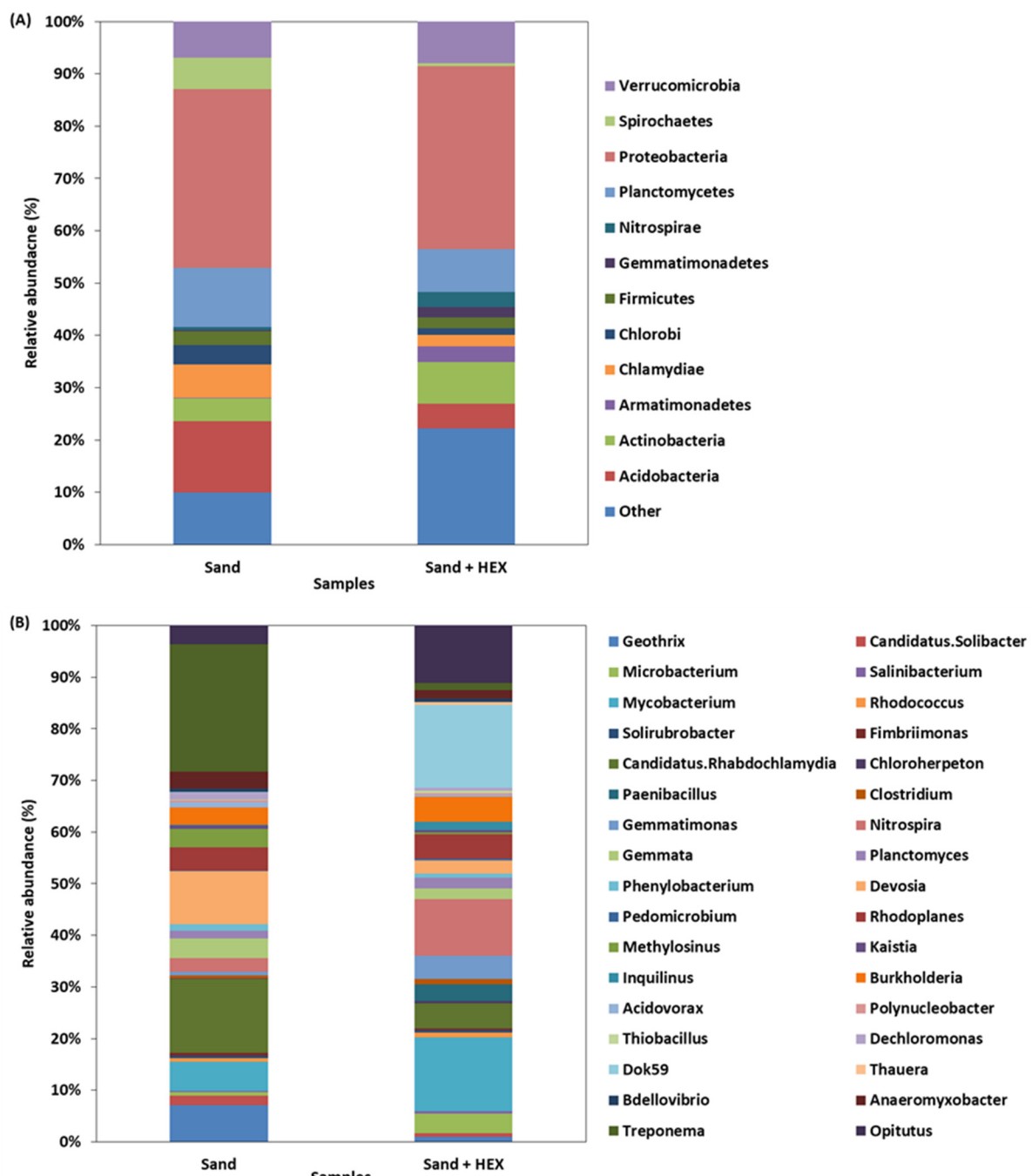

**Figure 7.** Diversity of the microbial community in the CIE treatment (**A**) phylum and (**B**) genus.

It has been reported that the phylum *Proteobacteria* contains lignin-degrading groups [41]. An additional study reported that *Actinobacteria* are also lignin-degrading organisms [42,43]. This information supported the performance of lignin–tannin removal in both reactors during this study, where the genera of the *Proteobacteria* (*Dok59*, *Burkholderia*, *Rhodoplanes*, *Devosia* and *Dechloromonas*) and the *Actinobacteria* (*Microbacterium*, *Salinibacterium*, *Mycobacterium*, *Rhodococcus* and *Solirubrobacter*) were responsible for the degradation. In addition, the presence of the genus *Nitrospira* in the DM-BR increased the removal of $NH_4^+$-N com-

pared to the C-BR. This genus has previously been shown to perform well in the oxidation of $NH_4^+$-N [44,45].

## 4. Conclusions

In this study, real coffee effluent was successfully treated using a DM-BR containing sand and HEX media under aerobic conditions. Coffee effluent is a high-strength effluent that contains non-degradable organic compounds under aerobic conditions. Changes in HRT had no significant effect on COD removal but did influence the removal of $NH_4^+$-N, colour and turbidity. The combination of HEX and sand media in the DM-BR showed improved removal performance at an HRT of 48 h compared to the sole application of sand media in the C-BR. The combination of media also encouraged the growth of various species of bacteria, with the majority belonging to the *Proteobacteria*.

**Author Contributions:** Conceptualization, H.A.H.; methodology, H.A.H. and D.S.A.S.; validation, H.A.H. and D.S.A.S.; formal analysis, D.S.A.S.; investigation, H.A.H. and D.S.A.S.; resources, H.A.H.; data curation, H.A.H. and D.S.A.S.; writing—original draft preparation, D.S.A.S.; writing—review and editing, H.A.H., M.H.M. and S.B.K.; visualization, H.A.H. and D.S.A.S.; supervision, H.A.H. and S.R.S.A.; project administration, H.A.H.; funding acquisition, H.A.H. All authors have read and agreed to the published version of the manuscript.

**Funding:** This research was financially supported by the Universiti Kebangsaan Malaysia (UKM) through grant number DIP-2021-008.

**Conflicts of Interest:** The authors declare no conflict of interest.

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
