# Peer review of "Potential of Using Dual-Media Biofilm Reactors as a Real Coffee Industrial Effluent Pre-Treatment"

_water, doi:10.3390/w14132025_

Round 1

Reviewer 1 Report

In this study, the dual media in biofilm reactor (DM-BR) for the treatment of real coffee effluent have been used. The performance of the DM-BR and C-BR in treating coffee effluent were investigated at different hydraulic retention times (HRTs) of 24, 48 and 72 hours and the degrading bacteria were identified. The result of this study is interesting, and the research fit for the scope of the Water journal. However, some explaination shold be further explained.

  1. Please compared the effect of DM-BR in treating coffee effluent with other reactors in the literature.
  2. The advantages and disadvantages for the various approachs applied to remove pollutant from wastewater, such as ZVI, UV, Feton, ect. are unclear in the introduction. Cost fator is not the only reason.
  3. In Table 1, the effluent is characterized, however the method and apparatus is not provided in the manuscript.
  4. Line 320-321, page 11, the sentence "The turbidity removal increased from 21.2% (24 hours HRT) to 32032.7% (48 hours HRT) but decreased to 23.8% as the HRT was increased to 72 hours."why? please clearly explain the result.

     5. The abstrat can be summized further and show important conclusions.

Reviewer 2 Report

The study investigated the feasibility of using a dual media biofilm reactor (DMBR) to treat the real coffee industrial effluent (CIE). In this study, the authors showed the performance of the DMBR at different HRTs regarding COD, ammonium, color, and tannin-lignin removal efficiencies. One minor contribution of this study to the literature is that the authors demonstrated the inefficiency of using DMBR to treat CIE although actually, this result can be seen without doing any experiments. The CIE is difficult to be degraded by aerobic digestion. In addition, instead of using the diluted raw CIE, the authors should use the effluent of the anaerobic digestion of the raw CIE as a feed to DMBR to improve the applicability of the proposed technology. In other words, a hybrid system using anaerobic digestion followed by DMBR would be more feasible. This publication might be not accepted to be published in the Water journal in the current form. Thorough modifications need to be made to improve the quality of the manuscript. The following can be considered:

1. Re-write the abstract to highlight the key findings and the novelty of this study

2. The authors should present the advantages and disadvantages of using the DMBR in this study because this configuration is similar to a moving bed bioreactor plus the sand filtration

3. Line 22-24, in the abstract, 'The combination of sand and HEX media…bacteria species’. What kinds of parameters the combination improved? Please, specify

4. In Table 3, please indicate the meaning of the standard deviation values. This is a basic error, please avoid it

5. There are plenty of grammar mistakes throughout the manuscript, making the readers confused. Please, avoid using long sentences. For example, line 290-293, 'the most effective...Table 3' and 'At HRTs of 24...performance'. Keep your writing succinct and clear

6. In figure 5, add more information to the Figure description. A1, A2, B1, B2, etc. mean what? If A2 is used to describe the A1 in detail, which part of A1 is described here? why the resolution of A1 and C1 is 200 micron, but that of B1 is 100 micron? So many basic errors are present in this manuscript. 

Reviewer 3 Report

Line 146 - it is essential to mention the granulometry of the used sand and if you can, the chemical composition, or mineralogical content.

Round 2

Reviewer 1 Report

The comments have been answered, so the manuscript can be accepted in the present form.

Reviewer 2 Report

The revised manuscript can be considered to be published in Water